# Effects of Frequent Smartphone Use on Sleep Problems in Children under 7 Years of Age in Korea: A 4-Year Longitudinal Study

**DOI:** 10.3390/ijerph191610252

**Published:** 2022-08-18

**Authors:** Sangha Lee, Sungju Kim, Sujin Yang, Yunmi Shin

**Affiliations:** 1Department of Psychiatry, Ajou University School of Medicine, Suwon 16500, Korea; 2Gwangju Smile Center for Crime Victims, Gwangju 61480, Korea

**Keywords:** screentime, sleep, children, longitudinal study, smartphone

## Abstract

The use of electronic screen devices has a negative effect on sleep. The purpose of this study is to longitudinally examine the effects of various screen use on sleep problems in children under 7 years of age. A total of 314 caregivers of children aged 4–7 years from three cities in Korea were recruited and responded to a self-administered questionnaire from 2017 to 2020. As a result of the analysis of the mixed model designed as a two-leveled structure, the use frequency of smartphones significantly predicted children’s sleep problems (β = 0.328, *p* < 0.001) compared to that of TV, PC, and tablet PC. In addition, the frequency of smartphone screen use showed a weak but significant correlation with bedtime resistance (r = 0.067, *p* = 0.009), sleep duration (r = 0.089, *p* <0.001), nighttime awakening (r = 0.066, *p* = 0.010), and daytime sleepiness (r = 0.102, *p* < 0.001). The results of this study suggest that screen time education in Korea should focus on smartphones above all else.

## 1. Introduction

In the past decade, the use of portable, screen-touch smart devices, such as smartphones and tablet personal computers (tablet PCs), as well as traditional electronic devices such as televisions (TVs) and personal computers (PCs), has increased not only in South Korea but also worldwide. Excessive screen time has a negative effect on physical and mental health, especially in children and adolescents [1,2,3]. Among the negative consequences, sleep is closely related to mental health. Sleep problems exacerbate depression, increase suicidal thoughts, and impair daytime function due to daytime sleepiness [4].

Poor sleep is a common and dangerous problem in adolescents, affecting 25–40% of children at some point during their development [5,6]. The spread of electronic devices such as tablet PCs and smartphones has been associated with children’s poor health [7,8,9]. TVs in the bedroom have been associated with later bedtimes, less time in bed, shorter sleep duration, and daytime sleepiness [10]. The habit of using smartphones until late at night resulted in bedtime procrastination and, as a result, sleep deprivation [11]. Korean children are used to using electronic devices from infancy. In Korea, 38% of toddlers under the age of 11 months start to use screen-based devices and consume screen-based media, spending approximately 2.4 h each using screens [12]. This usage behavior is believed to be related to sleep problems after reaching school age. It has been reported that bedtime smartphone use resulted in a substantial reduction in school day sleep duration among children aged 11–13 years [13].

TV exposure has been reported to be associated with sleep disorders in children. In a study of children aged 5–6 years, TV watching was associated with sleep–wake transition disorder and overall sleep disorder; in particular, passive TV exposure and watching TV programs for adults were strongly associated with sleep disorders [14]. A 7-year longitudinal study of 1864 children found that each year, the children’s sleep time decreased and the time they watched TV increased. The researchers argued that each 1 h per day increase in lifetime TV viewing was associated with a 7 min daily shorter sleep duration [15]. The association between computer use and sleep has mainly focused on relatively older adolescents [16]. A cross-national study using World Health Organization data analyzed data of 5402 adolescents from Finland, France, and Denmark, suggesting that more computer use is associated with shorter sleep duration and negative emotional symptoms [16].

According to a recent review with meta-analysis, electronic device use and child sleep outcomes were negatively correlated. In addition to sleep duration, a range of sleep indicators were associated with daily smartphone and tablet PC use, including bedtime resistance, overnight awakenings, sleep onset latency, and risk of sleep disturbance [17]. Night-time technology use was also associated with functional impact, including increased sedentary behaviors prior to bed, subjective poor sleep quality, greater caffeine consumption, falling asleep in school, and increased daytime sleepiness. In addition, according to our previous study [18], excessive smartphone use was related to shorter total sleep time as well as severely decreased sleep quality in younger children.

However, another study has reported that traditional electronic devices such as TVs have a stronger negative correlation with sleep compared to mobile screens. To the best of our knowledge, there are few studies comparing the effects of different electronic devices on sleep problems. In addition, most studies are carried out as cross-sectional studies, reporting only the correlation between electronic device screen time and sleep [17]. It is difficult to ascertain how the frequency of electronic device use will affect children’s sleep in the future.

In order to address this gap, the current study aims to longitudinally examine the effects of screen use frequency on children’s sleep and to confirm which sleep problems are specifically related.

## 2. Materials and Methods

### 2.1. Participants

The Kids Cohort for Understanding of Internet Addiction Risk Factors in Early Childhood (K-CURE) study is the first long-term observational prospective cohort study to investigate causal factors associated with internet-related disorders in Korea. This study was conducted as a part of the K-CURE study and utilized data from wave 4 to wave 7. We invited caregivers (mostly mothers) to participate in a self-administered survey on children’s usage of various media platforms, as well as the mental health of children and their caregivers, including sleep health. The caregivers made a voluntary visit to a community center for child mental health located in Suwon, Sungnam, or Goyang, all of which are major cities in the most populous province of Korea. A total of 340 caregivers of children between the ages of 5 and 7 were recruited from the general population at baseline (T1). The criterion for participant enrollment was having elementary school or preschool children. The authors verified whether they owned each device. Children with serious developmental disabilities, such as autism or intellectual disability, were excluded from the study. The multi-level data used in this study were divided into level 1 (time) and level 2 (individual). Level 1 variables consisted of a total of 4-time points (T1–T4). Since there were some drop-outs and missing values every year, in this study, data from 312 people who finally remained in T4 with no missing values were used (Figure 1).

### 2.2. Measurement of Screen Use Frequency

Respondents were asked to report whether they had the following media devices: TV, PC, including laptops, tablet PC, and smartphone. In addition, respondents were asked how many days per week, on average, their children used media devices over the course, on average, of the month.

### 2.3. Measurement of Sleep

Children’s sleep duration and sleep habits were assessed using a parent-reported questionnaire. To assess sleep duration, parents were asked to reply to the following question: “How long does your child sleep per day, excluding naps?”. Sleep characteristics of the children were assessed using the Children’s Sleep Habits Questionnaire (CSHQ) [19]. The CSHQ represents a parent-reported sleep-screening instrument that has been used in a number of studies to examine sleep behavior in young children [20,21,22]. Items on the CSHQ were rated on a three-point Likert scale ranging from rarely (0–1 time per week), sometimes (2–4 times per week), and usually (5–7 times per week) to reflect children’s sleep habits over a recent typical week. Bedtime resistance, sleep onset delay, sleep duration, sleep anxiety, nighttime awakenings, daytime sleepiness, parasomnias, and sleep-disordered breathing were among the 33 items of the CSHQ divided into eight subscales. The sum of all scored items yielded a total sleep disturbance score ranging from 33 to 99, with a higher score indicating greater sleep disturbances. A screening cut-off score of >41 reflects clinically significant sleep disturbances [19]. However, using the original CSHQ cut-off score, too many children fall into the group with sleep problems. Based on the research that about a quarter of preschool children experience sleep disturbance [23], the cut-off in this study was adjusted to >48. This problem was also pointed out in previous studies, and similarly, the cut-off score was adjusted to 48 or 52 [24,25]. The internal consistency and test–retest reliability of the CSHQ was 0.70 and 0.79, respectively [19].

### 2.4. Data Analyses

Statistical analyses were performed using the Jamovi program (ver. 2.2.5). In this study, the change in frequency of media use and sleep problem score of children over time and the factors influencing these changes were investigated. We used data in the form of variables organized as structures nested within each individual’s data entry for each time point. The multi-level model is known to be suitable for analyzing individual changes by reflecting the data structure of multiple levels inherent in the individual’s measurements [26]. Therefore, this study sought to analyze by applying a two-level multi-layered model that reflects the data structure consisting of two levels: the individual (level 2) and the measurement value by time (level 1) inherent in the individual. Statistical significance was set at *p* < 0.05. The multi-level data used in this study were divided into level 1 (time) and level 2 (individual). The level 1 variables consisted of a total of 4-time points (T1–T4), and the level 2 variables consisted of the frequency of use of each device and CSHQ score. Since there was a significant difference in the frequency of tablet PC use between males and females at baseline, gender was used as a control variable. The research model for each level was as follows:

Level 1 (time)
Y_*ij*_ = *β*_0*i*_ + *β*_1*i*_*X_j_* + … + *β*_0*ij*_ + *r_ij_* ~ *N*(0, *σ*2)

Level 2 (individual)
*β*_0*j*_ = *γ*_0*i*_ + *u*_0*j*_,
…,
*β*_6*j*_ = *γ*_60_ + *u*_6*j*_

In the above equation, Y*_ij_* means the sleep problem score of subject j measured at time *i*. *X* is the measurement time, and *r_ij_* is the residual. *γ*_1*j*_~*γ*_6*j*_ means the two-level variables: TV use frequency, PC use frequency, tablet use frequency, smartphone use frequency, gender, and age. *u_ij_* stands for a random effect. Missing value treatment uses list-wise deletion, the most common method for handling missing values so that even in one case, missing items exist. In this case, the entire data in the row were deleted and excluded from the analysis, i.e., only the data without missing values in the main variables among the data for which follow-up was completed from T1 to T4 were analyzed.

### 2.5. Ethics

This study was approved by the Institutional Review Board at the Ajou University School of Medicine (AJIRB-SBR-SUR-14-378). Informed consent was obtained from all participants when they were enrolled.

## 3. Results

### 3.1. Demographic Information

Table 1 presents baseline demographic information, device usage frequency, total sleep time, CSHQ score, and the percentage of participants at high risk for sleep problems. Of the total 312 participants, 155 (49.7%) were male, and 157 (50.3%) were female. To assess if there was a difference in baseline between the sexes, the t-test and chi-square test were conducted. The average age of the participants was 5.38 years, and school-age children accounted for 10.6% of the total participants.

Participants watched TV on average 4.65 (±2.41) days a week, using PC 0.96 (±1.85) days, tablet PC use 1.93 (±2.56) days, and smartphone use 2.90 (±2.52) days. There was no significant difference in the frequency of use of each device according to age. In addition, the frequency of use of tablet PCs was significantly higher in females than in males (mean difference = 0.68, *p* = 0.018).

### 3.2. Yearly Changes and Trajectories of the Variables

The weekly average TV use frequency decreased to 4.18 days in T2, increased to 4.83 days in T3, and decreased again to 4.38 days in T4. The frequency of PC use increased to 1.03 days in T2, 2.79 days in T3, and decreased to 2.47 days in T4. The usage frequency of Tablet PC decreased to 1.87 days in T2, increased to 2.96 days in T3, and decreased to 2.85 days in T4. Meanwhile, the frequency of smartphone use was 3.25 days for T2, 3.73 days for T3, and 4.38 days for T4 (Table 2).

Total sleep time continued to decrease on an annual basis. Children who slept an average of 598 min at baseline (T1) steadily decreased to 583 min at T2, 580 min at T3, and 555 min at T4.

The CSHQ score also decreased gradually each year. The CSHQ score, which was an average of 44.4 in T1, decreased to 44.3 in T2, 42.7 in T3, and 41.8 in T4. The number of high-risk groups exceeding 48 points in CSHQ also decreased from 67 in T1 to 45 in T3 and 34 in T4 (Figure 2).

### 3.3. Factors That Longitudinally Influence Sleep Problems

Table 3 presents the result of linear multi-level regression. Intraclass correlation (ICC) values calculated to examine the proportion of variance over time among the total variance of CSHQ score showed that the time variance accounted for approximately 3.1% of the total variance, and the individual variance accounted for approximately 96.9%.

To assess which device variables are related to sleep problems over time, the intra-subject model, i.e., the random coefficient model, was analyzed. As seen in Table 4, the frequency of smartphone use affected the CSHQ score, and the higher the frequency of smartphone use, the higher the CSHQ score (e = 0.328, 95% confidence interval (CI): 0.212–0.443, *p* < 0.001).

### 3.4. Correlation between Device Use and Sleep Problems

Table 5 presents correlations between the examined variables, including subscales of CSHQ. The frequency of smartphone use was positively and significantly correlated with the frequency of TV and PC use (r = 0.150, *p* < 0.001; r = 0.100, *p* < 0.001, respectively), but there was no correlation with the use frequency of tablet PC (r = −0.092, *p* = 0.999). There was a weak correlation between the total CSHQ score and the frequency of smartphone use (r = 0.123, *p* < 0.001) and TV (r = 0.051, *p* = 0.037). The bedtime resistance was positively correlated to TV and smartphone use (r = 0.075, *p* = 0.004; r = 0.067, *p* = 0.009, respectively), but the bedtime anxiety was only correlated with TV use (r = 0.086, *p* = 0.001). Smartphone use, in addition to bedtime resistance, was weakly correlated with sleep onset (r = 0.055, *p* = 0.026), sleep duration (r = 0.089, *p* < 0.001), nighttime awakening (r = 0.066, *p* = 0.01), and daytime sleepiness (r = 0.102), *p* < 0.001). PC and tablet PC use demonstrated no significant correlation with any of the CSHQ subscales.

## 4. Discussion

This study aimed to identify factors affecting children’s sleep problems by longitudinally analyzing the screen use frequency and sleep patterns of the same participants over the course of 4 years. In contrast to the previous review study [10], most of the effects of electronic device use on sleep problems were not significant, and the frequency of smartphone use was mainly related to the decrease in children’s sleep time.

The key findings and implications of this study are as follows. Comparing the screen time of children over the past 4 years, the frequency of TV use did not change significantly from year to year. In contrast, however, the frequency of use of PC and tablets PC increased significantly at T3 but decreased to an insignificant level at T4, and the frequency of smartphone use increased steadily every year. In Korea, since only one TV is usually owned by a household [27], families share the same TV; therefore, the average frequency of use of the device for a week is highly likely to reflect the average frequency of use of the household. Furthermore, the widespread presence of smart TVs in Korea has increased significantly over the past 5 years. The annual survey on media usage behavior in Korea reveals that the smart TV ownership rate has more than doubled from 15.9% in 2017 to 33.6% in 2020 [27]. Smart TVs are believed to have functioned as smart devices, unlike traditional TVs, since they can use the contents of smartphones or tablets [28]. In contrast, however, other devices are relatively personalized; as such, it is believed that as the number of school-age children increases, the use of mobile games and social networking services, as well as online learning, will increase. Among the study participants, the proportion of school-aged children was only 10.6% at the baseline and increased to 84.2% at T3, which explains the surge in overall screen time at T3. Meanwhile, smartphones are more likely to be owned by children than any other device [29], and the frequency of use seems to have continuously increased, reflecting the fact that it remains difficult for parents to control their use. Since smartphones share functions with PCs and tablets in many areas [13], it is highly likely that smartphones have replaced the time spent on other devices at T4 [30,31] when all participants reach school age.

Year by year, participants’ total sleep time decreased, as did the total CSHQ score. The decrease in total sleep time as children get older seems to reflect a natural process of growth [32]. In addition, since co-sleeping is very common in Korea before the age of 10 [33], the scores on questions such as “Needs parent in room to sleep”, “Afraid of sleeping alone”, and “Afraid of sleeping in the dark” among the CSHQ questions are expected to decrease as children grow, and this may have contributed to the gradual decrease in total CSHQ score. As the total CSHQ score decreased, the proportion of children exceeding the cut-off point (48) naturally decreased every year. It may be suggested that children who continue to be at high risk after having reached school age may experience sleep problems due to poor sleep hygiene or inadequate sleep habits.

The association between screen time and sleep problems in traditional electronic devices has been reported for a long time [10,11,14,17]; however, this study did not fully support such previously identified findings. As mentioned above, since the time variance accounted for only 3.1%, it can be seen that although the variance over time is significant, it is much more affected by individual differences, such as the frequency with which devices are used. Multi-level analyses of the level-1 variable with time (intra-individual variable) and the level-2 variable with the individual revealed that the frequency of TV and PC, as well as tablet use, did not significantly predict sleep problems in T4, but only the frequency of smartphone use predicted the total sleep problems score. In the subscales of CSHQ, there were significant positive correlations with smartphone usage frequency and bedtime resistance, sleep onset, sleep duration, nighttime awakening, and daytime sleepiness. The frequency of TV use also revealed a weak correlation with bedtime resistance and sleep anxiety, but most of the indicators had no correlation with the frequency of use of other devices. These results suggest that smartphones may play a more important role than other devices in the sleep problems of Korean children [31]. As previously mentioned, TVs and PCs are not owned by individual family members; thus, parental control is relatively easy, and tablets share most functions with smartphones. Smartphones are the most ergonomic devices and are easy to use in bed before a child goes to bed [34].

Bedtime resistance, sleep onset, sleep duration, and daytime sleepiness are all positively correlated subscales, and they are likely to be related to the habit of using smartphones until late at night. The use of smartphones at night delays the onset of sleep by suppressing the secretion of melatonin [35,36], which plays an important role in initiating sleep [37]. Unfortunately, children who have to wake up in time for school inevitably lack sleep. Consequently, daytime sleepiness is an unavoidable consequence. From another perspective, this behavior can be interpreted as bedtime procrastination in childhood. Bedtime procrastination refers to behaviors whereby an individual goes to bed later than the intended time despite the absence of external factors [38]. It can become a habit during childhood and may be health-threatening. According to a related study, students who engage in bedtime procrastination have a higher level of stress and a higher incidence of physical illness [39].

This study has some limitations. First, it was not possible to accurately assess the time period during which children actually use smartphones. Assuming that the frequency of smartphone use reflects habitual use, dependence on the smartphone should be confirmed. Second, it was not possible to investigate which devices were in the children’s bedroom. Previous studies have reported that having a TV and PC in the bedroom has a significant effect on the quality and quantity of sleep, but this aspect was not assessed in the current study. Third, we used frequency of device use rather than screen time. In several studies involving children, screen time is assessed based on parental reports, which is a subjective third-party report that relies on memory; therefore, there is a risk that it may differ from the children’s actual screen time [40]. In particular, in the case of background TV, some parents include it in the children’s screen time while others do not [41,42,43]. Parents may not be able to accurately grasp the actual screen time of smartphone use in their children. Moreover, when both parents work, this time distortion may be particularly large. Therefore, the researchers tried to understand the usage behavior by focusing on the frequency with a relatively low possibility of distortion. It is expected that the association between screen time and sleep problems can be predicted more precisely using objective methods that can assess device usage time, such as smartphone usage trackers, in the future. Finally, the scale for evaluating sleep problems was limited to total sleep time and CSHQ. Although CSHQ probes sleep problems based on various aspects, there remain limitations associated with the parental self-report questionnaire. For a more accurate evaluation of sleep problems, it is necessary to record a sleep diary for at least 2 weeks and supplement it with objective devices, such as devices using actigraphy.

Despite some limitations, this study is significant in that it is the first longitudinal study to assess the effects of media device use on sleep among children in Korea. In this study, following the study comparing the sleep scores of children who use smartphones with those who do not use smartphones in the past K-CURE study [44], the effects of other devices were analyzed together, and the results of a longitudinal study were presented. Further studies using more objective and precise measurements of children’s sleep and screen time will be needed.

## 5. Conclusions

These results lead to a few key implications. As a factor influencing children’s sleep problems in Korea, smartphones had the greatest influence, as compared to other devices. Most children seem to have fewer sleep problems as they grow up, but smartphone usage increases significantly as they reach school age, which delays the onset of sleep and shortens their sleep period.

## Figures and Tables

**Figure 1 ijerph-19-10252-f001:**
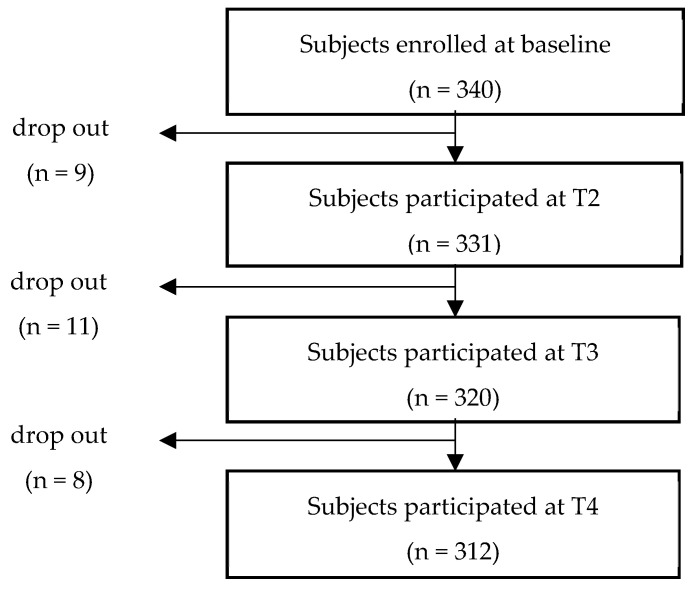
The flow of drop-outs each year.

**Figure 2 ijerph-19-10252-f002:**
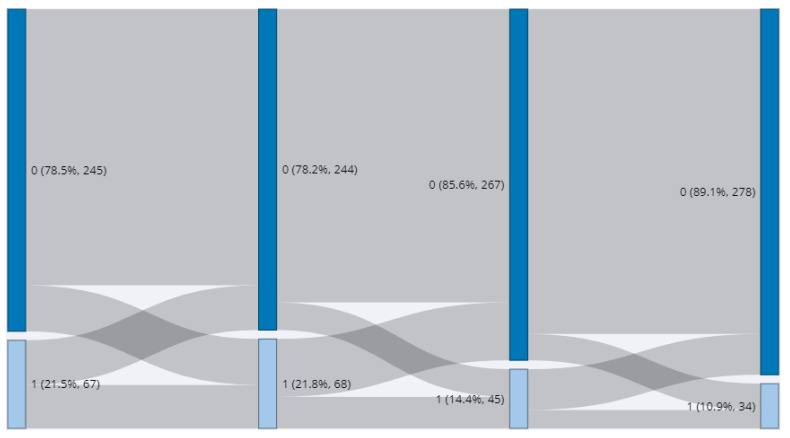
Year-by-year trajectory of a high-risk group for sleep problem (1 = high risk).

**Table 1 ijerph-19-10252-t001:** Demographics of participants at baseline (n = 312).

Variables	Age				F	*p*
	4	5	6	7		
Number of participants	49	129	101	33		
Sex = female (n, %)	25 (51.0%)	69 (53.5%)	53 (52.5%)	10 (30.3%)		
Media use frequency (days/week)	mean (SD)					
TV	4.98 (2.38)	4.61 (2.40)	4.65 (2.39)	4.27 (2.54)	0.56	0.642
PC	0.778 (1.60)	0.71 (1.64)	1.15 (2.09)	1.64 (2.06)	2.511	0.063
Tablet	2.57 (2.91)	1.64 (2.29)	2.12 (2.71)	1.58 (2.41)	1.824	0.147
Smartphone	2.71 (2.55)	2.96 (2.40)	2.73 (2.63)	3.48 (2.59)	0.802	0.496
Sleep	mean (SD) or n (%)					
Total sleep time (minutes)	590 (43.8)	597 (35.1)	603 (37.8)	598 (34.7)	1.232	0.302
CSHQ score	44.5 (6.48)	45.0 (5.20)	43.6 (5.76)	44.2 (5.09)	1.236	0.3
High-risk group	11 (22.4%)	28 (21.7%)	20 (19.8%)	8 (24.2%)		

**Table 2 ijerph-19-10252-t002:** Year-by-year changes in screen frequencies and sleep problems.

Year		2017	2018	2019	2020
Age (n)	4	49	-	-	-
	5	129	49	-	-
	6	101	129	49	-
	7	33	101	129	49
	8	-	33	101	129
	9	-	-	33	101
	10	-	-	-	33
Frequency (days)	TV	4.65 (2.41)	4.18 (2.44)	4.83 (2.54)	4.38 (2.54)
	PC	0.96 (1.85)	1.03 (1.75)	2.79 (2.72)	2.47 (2.67)
	Tablet	1.93 (2.56)	1.87 (2.43)	2.96 (2.88)	2.85 (2.90)
	Smartphone	2.90(2.52)	3.25 (2.61)	3.73 (3.00)	4.38 (3.00)
Total sleep time (minutes)		598 (37.5)	583 (32.5)	580 (40.9)	555 (38.4)
CSHQ total (mean, SD)		44.4 (5.59)	44.3 (5.55)	42.7 (5.70)	41.8 (5.86)
CSHQ high-risk group (n, %)		217 (69.6%)	211 (67.6%)	176 (56.4%)	148 (47.4%)

**Table 3 ijerph-19-10252-t003:** One-Way ANOVA (Welch’s) and Games-Howell post hoc test of CSHQ score.

		F	df1	df2	*p*
CSHQ		15.4	3	691	<0.001
**Time**		**T1**	**T2**	**T3**	**T4**
T1	Mean difference	-	0.0994	1.72 ***	2.603 ***
	*p*-value	-	0.996	<0.001	<0.001
T2	Mean difference		-	1.63 **	2.503 ***
	*p*-value		-	0.002	<0.001
T3	Mean difference			-	0.878
	*p*-value			-	0.230
T4	Mean difference				-
	*p*-value				-

** significant at *p* < 0.01 level; *** significant at *p* < 0.001 level.

**Table 4 ijerph-19-10252-t004:** Fixed effects parameter estimates.

Effect	Estimate	SE	95% Confidence Interval		df	t	*p*
			Lower	Upper			
(Intercept)	43.222	0.529	42.185	44.259	2.25	81.678	<0.001
TV_frequency	0.059	0.066	−0.071	0.189	1239.47	0.896	0.371
PC_frequency	−0.042	0.070	−0.179	0.095	1227.61	−0.603	0.547
Tablet_PC frequency	−0.004	0.060	−0.121	0.113	1236.36	−0.069	0.945
Smartphone frequency	0.328	0.059	0.212	0.443	1236.98	5.559	<0.001
Sex	0.355	0.321	−0.275	0.984	1237.83	1.105	0.269
Age	−0.352	0.176	−0.696	−0.007	39.64	−2.001	0.05

**Table 5 ijerph-19-10252-t005:** Correlation between variables and CSHQ subscale scores.

			1	2	3	4	5	6	7	8	9	10	11	12
1	TV	r	-											
		*p*	-											
2	PC	r	−0.068	-										
		*p*	0.992	-										
3	Tablet PC	r	0.04	0.023	-									
		*p*	0.079	0.212	-									
4	Smartphone	r	0.15 ***	0.1 ***	−0.092	-								
		*p*	<0.001	<0.001	0.999	-								
5	CSHQ_total	r	0.051 *	−0.072	−0.045	0.123 ***	-							
		*p*	0.037	0.995	0.945	<0.001	-							
6	CSHQ_BR	r	0.075 **	−0.108	−0.067	0.067 **	0.68 ***	-						
		*p*	0.004	1.000	0.991	0.009	<0.001	-						
7	CSHQ_SOD	r	−0.051	0.026	0.035	0.055 *	0.402 ***	0.149 ***	-					
		*p*	0.965	0.181	0.107	0.026	<0.001	<0.001	-					
8	CSHQ_SD	r	−0.053	0.024	−0.041	0.089 ***	0.491 ***	0.136 ***	0.421 ***	-				
		*p*	0.97	0.199	0.925	<0.001	<0.001	<0.001	<0.001	-				
9	CSHQ_ANX	r	0.086 **	−0.105	−0.037	0.013	0.593 ***	0.796 ***	0.123 ***	0.052 *	-			
		*p*	0.001	1.000	0.907	0.320	<0.001	<0.001	<0.001	0.033	-			
10	CSHQ_NA	r	0.025	−0.069	−0.054	0.066 *	0.448 ***	0.222 ***	0.082 **	0.075 **	0.275 ***	-		
		*p*	0.188	0.993	0.972	0.010	<0.001	<0.001	0.002	0.004	<0.001	-		
11	CSHQ_PS	r	0.036	−0.009	0.002	0.006	0.455 ***	0.127 ***	0.108 ***	0.054 *	0.16 ***	0.256 ***	-	
		*p*	0.104	0.631	0.466	0.417	<0.001	<0.001	<0.001	0.029	<0.001	<0.001	-	
12	CSHQ_SDB	r	−0.002	0.028	−0.077	0.038	0.21 ***	0.026	0.05 *	0.035	0.022	0.087 **	0.217 ***	-
		*p*	0.534	0.160	0.997	0.089	< .001	0.176	0.04	0.108	0.215	0.001	<0.001	-
13	CSHQ_DS	r	0.026	−0.027	0.017	0.102 ***	0.673 ***	0.138 ***	0.205 ***	0.363 ***	0.06 *	0.099 ***	0.168 ***	0.114 ***
		*p*	0.175	0.827	0.280	<0.001	<0.001	<0.001	<0.001	<0.001	0.018	<0.001	<0.001	<0.001

* significant at *p* < 0.05 levels; ** significant at *p* < 0.01 level; *** significant at *p* < 0.001 level. Abbreviations: CSHQ—Children’s Sleep Habits Questionnaire; TV—television; PC—personal computer; BR—bedtime resistance; SOD—sleep onset delay; CSHQ_SD—sleep duration; CSHQ_ANX—sleep anxiety; NA—nighttime awakenings; PS—parasomnias; SDB—sleep-disordered breathing; DS—daytime sleepiness.

## Data Availability

The datasets during and/or analyzed during the current study are available from the corresponding author on reasonable request.

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
