# Peer review of "Effects of Frequent Smartphone Use on Sleep Problems in Children under 7 Years of Age in Korea: A 4-Year Longitudinal Study"

_ijerph, 2022, doi:10.3390/ijerph191610252_

Round 1

Reviewer 1 Report

The main limitation of this study is that included participants are children with internet-related disorder or mental health problem, not general pediatric population. You should state about that.

How many children drop-out or missing every year? Please add the flow chart. If the data of dropped-out patients or missing values is deleted in the analyses, selection bias occur.

The proportion of high risk for sleep problem is too high in this study. You should consider and adjust other causal factors associated with sleep problem (stress level, time to get home from academy..).

In Table 1, why did you compare the demographics of participants according to sex? It would be better to compare the demographics according to the age.

In Table 2, you should show the changes of the variables according to the age.

Please add the reference about the screening cutoff score (> 41) of sleep disturbances.

Please delete the sentences (in Line 149-151).

Reviewer 2 Report

This work showed an interesting investigation about the effects of various screen use on sleep problems in children under 7 years of age. The research is well designed and performed, and the manuscript is clearly presented. However, I have a number of further comments to improve the manuscript:

Introduction:

·    It’s better to mention and describe the statistical (analysis) method that used in this work. Then, explain why this work used it based on the literature review.

·   Add one paragraph for organisation of the paper at the end of introduction section.

Materials and Methods:

·    How long the sleep duration (or days) of each subject that used in this work?

·    All equation parameters/variables should be explained and numbered.

Results:

·    Figure 1 and related argument: How to read the figure and the value of CSHQ, as like “0 (30.4 %, 95)”. It should be commented on in the manuscript.

Discussion:

·      It is better to report the performance of the proposed system that compared with the other system in to justify its advantage and value, might present to be in a summary table.

·    Conclusion:
Basic sections have been covered, future section can be added.

Round 2

Reviewer 1 Report

No other comments.